# Fire Safety Index for High-Rise Buildings in the Emirate of Sharjah, UAE

**Musab Omar \***, **Abdelgadir Mahmoud and Sa'ardin Bin Abdul Aziz**

Razak Faculty of Technology and Informatics, Universiti Teknologi Malaysia, Kuala Lumpur 54100, Malaysia
* Correspondence: alkhaldy70@hotmail.com

**Abstract:** The purpose of this paper was to develop a fire index system for measuring the compliance of high-rise residential buildings with fire requirements in the Emirate of Sharjah, and also to develop an index system for measuring the fire response efficiency, which is linked the two indexes, and the higher the compliance rate, the greater chance of a successful response. The two systems depend on the automation of the firefighting system management processes using the techniques of the fourth industrial revolution, and they were developed based on consultation with subject matter experts in the field and used multiple methods, such as the analytic hierarchy process, failure mode effect, criticality analysis, and Delphi techniques. The main criteria of the indexes were identified as the fire risk assessment, fire suppression system, fire accident management, fire alarm system, fire extinguisher, employees, residents, service rooms, lifts, gas connections, waste, housekeeping, and evacuation facilities. Each main criterium was detailed in the sub-criteria and weighted to achieve the index for each sub-criteria based on the fire legislation in UAE, the fire response index (developed based on the high-rise building fire index), the category of the building in terms of floor numbers, and the distance between building and the fire station. The two index systems can contribute to improving emergency preparedness in high-rise residential buildings in the Emirate of Sharjah and are also considered as measurement indexes for compliance with fire requirements in the Emirate of Sharjah.

**Keywords:** fire factor; residential buildings; fire accidents





## 1. Introduction

The Emirate of Sharjah is the third in the ranking of the UAE in terms of the number of high-rise residential buildings, and based on the analysis of accidents in the previous nine years (from 2013 to 2020), an increase in fire accidents in residential buildings was observed, which confirms the need to improve the fire management system in high-rise residential buildings and to develop an index system that monitors compliance with fire requirements in high-rise residential buildings. This would help to determine the required minimum requirements. Compliance falls under one of the variables that affect the success of the fire response process. A system for measuring response efficiency based on compliance rate, number of floors, and geographical location would contribute to improving the fire management system in the Emirate of Sharjah.

A building's functional diversification makes fire prevention is more complex. For the layers and sizes of high-rise buildings, a high concentration of people and property make the firefighting and evacuation operations very difficult when a fire occurs [1]. The most crucial aspect of a building's safety in the face of a fire is the possibility of safe escape. An important precondition is that a building's fire safety facilities enable independent and adequate fire response performances by the building's occupants [2]. Currently, a country such as China may opt to use the fourth Industrial Revolution (IR 4.0) for fire management. "Industry 4.0" is a term that was coined in Germany in 2011 to describe technology that fits into the design principles of interconnectivity, information transparency,

technical assistance, and decentralized decisions [3]. The intelligent evacuation guidance system (IEGS) is a new concept and product in China that uses an intelligent inducing algorithm to obtain dynamic evacuation routes and improve evacuation efficiency [4]. The accurate prediction of occupant evacuation is important in the evaluation of performance and risk analyses of buildings where large numbers of people may gather or where an emergency evacuation may be needed. Owing to the importance of life safety, especially as performance-based fire codes are adopted, the prediction of occupant evacuation has been one of the most critical parts of fire risk analysis. As a result, a large number of evacuation models have been proposed to meet the demand [5]. Fire safety can be defined as a set of practices for preventing fires, managing fire growth, and managing a fire's effects—either intentionally or unintentionally—while keeping the resulting losses at an acceptable level [6]. The fire safety levels of existing buildings decrease over time. In order to ensure the safety of buildings, hardware upgrades and fire safety management measures are essential [7].

Fire risk indexing (FRI) methods are heuristic models of fire safety. Heuristics are procedures that provide a practical approach to solving problems in the absence of a formal underlying physical theory [8], and they are typically defined as efficient rules or procedures for converting complex problems into simpler ones [9]. Heuristic methods refer to problem solving that employs a practical method that is not guaranteed to be optimal or perfect, but is instead considered sufficient for reaching an immediate goal [10]. In the context of fire safety in buildings, the objective of a heuristic method is typically to make decisions about the fire safety measures that should be included within a building, often with the aim of deploying limited resources to a maximum effect. These fire safety evaluation systems have been referred to by various names such as risk ranking, index systems, scoring, point schemes, and numerical grading [11].

In recent years, the number of high-rise building fires has remained high, and such fires pose serious threats to people's lives and property [12]. High-rise buildings have a greater risk of fire according to their features, such as a great height, complex structure, and diverse functions [13]. Due to the weakness of fire prevention capabilities in high-rise buildings, once a fire begins, human life, animal life, health, and property are all threatened [14]. With rapid economic development and urbanization, more and more high-rise residential buildings are built, particularly in densely populated areas, and the fire safety of high-rise buildings has attracted attention over the years due to its unique challenges, such as long evacuation times and distances, smoke movement and control, and fire department accessibility [15]. Several studies exist in the literature that have tried to investigate high-rise building fires either experimentally or numerically [16]. Fire risk assessment is a basic way to reduce and control fires in high-rise buildings [17]. The development of early warning indicators to prevent major accidents—to 'build safety'—should rest on a sound theoretical foundation that includes basic concepts, primary perspectives, and past developments, as well as an overview of the present status of fire safety and ongoing research [18].

## 2. Methods

To achieve the purpose of this paper (to develop a fire index to contribute to improving fire prevention in high-rise buildings in the Emirate of Sharjah), a literature review was carried out to identify the factors affecting the fire prevention management systems in residential buildings such as high-rise buildings (HRBs). Through the use of Delphi techniques, we presented the factors from the literature review to 16 subject matter experts in fire management systems, each from different stakeholder groups, such as the Sharjah Prevention and Safety Authority, the Sharjah Civil Defense Authority, fire maintenance and installation contractors, fire equipment agents and distributers, and fire office consultants, in order to identify the criteria affecting the fire prevention management systems in HRBs in the Emirate of Sharjah. Based on the analytic hierarchy process (AHP), which is able to assess the overall fire safety level of high-rise buildings in the Emirate of Sharjah in terms of the fire protection measures implemented and the use of the failure mode effect and

criticality analysis (FMECA), we proposed that the fire index use the methodology as shown in Figure 1. Methods of the research). The analytic hierarchy process (AHP) is a widely used model for dealing with multi-criteria decision-making (MCDM) problems. MCDM represents a sub-discipline of operations research that deals with solving decision-making problems that involve multiple, usually conflicting [19], elements. The most important element of MCDM is that it was developed and implemented an analytic hierarchy process (AHP) to determine the weights of indicators that provide extra strength and credibility for the results [20].

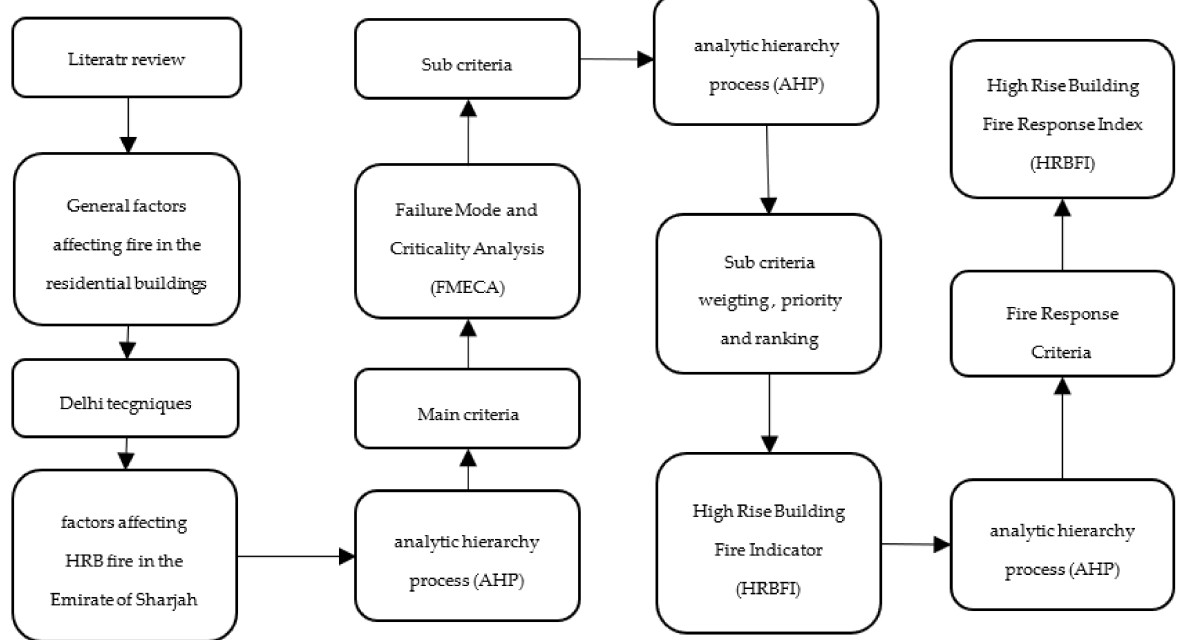

**Figure 1.** Methods of the research.

### 3. Results

From the literature review, 19 factors affecting fires in residential building were identified, and we classified them into three categories: management factors, human factors, and technical factors, such as fire regulations, fire enforcement regulations, fire research and development, facilities management and policies, fire data analysis/availability, accident investigations, public/contractor attitudes, rescue speed, staff assignments, human behavior, fire training, fire knowledge, fire culture of society, fire technology, lack of/improper maintenance, fire equipment, building design, combustible materials, and climate change. The factors were identified after we reviewed papers [21–35], and not all the factors were applicable to the Emirate of Sharjah. It appeared that after conducting three round of Delphi techniques, the main elements affecting high-rise residential buildings were the identifies and representations provided to the subject matter experts through the analytic hierarchy process (AHP), and the outcomes are shown in Table 1. The main elements affecting fire safety in HRBs.

It is shown that fire risk assessment is the top element affecting fires in HRBs, accounting for 22% due to the significant changes in the materials used for the construction and operation phases of HRBs, and failure to identify fire risk will contribute to increasing the possibility of a fire. Fire suppression systems ranked second with a percentage of 18%, and they are capable of assisting with the protection of a building through automatically dealing with fires. Fire accident management ranked third, with 16%, as the reporting of fire, investigations, and analyses play critical roles in identifying the gaps which require immediate correction, as do preventative corrections to avoid occurrences. Fire alarm systems accounted for 16%, and they play a role in the early notification of fires. Fire extinguishers accounted for 7%, employees working in HRBs accounted for 7%, residents

accounted for 7%, and service rooms, lifts, and gas connections accounted counted 4%, while waste, housekeeping, and evacuation facilities accounted for 6%. Through the Failure Mode Effect and Criticality Analysis (FMECA), each element was evaluated to detail the possible failures, and the outcomes of the evaluation concluded with the establishment of sub-criteria through the analytic hierarchy process (AHP), with weights determined as shown in Table 2. Sub-criteria weighting.

**Table 1.** The main elements affecting fire safety in HRBs.

| Criteria | Weights | Absolute Error | Ranking |
|---|---|---|---|
| Lambda = 9.429 (CR = 0.037) | | | |
| Fire risk assessment | 0.223 | 0.048 | 1 |
| Fire suppression system | 0.188 | 0.060 | 2 |
| Fire accident management | 0.166 | 0.053 | 3 |
| Fire alarm system | 0.128 | 0.040 | 4 |
| Fire extinguisher | 0.076 | 0.034 | 5 |
| Employees | 0.068 | 0.015 | 6 |
| Residents | 0.063 | 0.013 | 7 |
| Service rooms and lift and gas connections | 0.046 | 0.024 | 8 |
| Waste, housekeeping, and evacuation facilities | 0.043 | 0.012 | 9 |

**Table 2.** Sub-criteria weighting.

| Criteria | Weight | Sub-Criteria | Weight | Ranking |
|---|---|---|---|---|
| Fire risk assessment | 0.223 | – Competent fire risk professional in place | 0.353 | 1 |
| | | – Fire risk assessment carried out | 0.265 | 2 |
| | | – The control of fire risk assessment is effective, monitored, and reviewed | 0.202 | 3 |
| | | – Implement a fire safety management system | 0.180 | 4 |
| Fire suppression system | 0.188 | – All suppression systems are connected to a 24/7 Aman early warning system | 0.225 | 1 |
| | | – All pumps are energized 24/7 | 0.151 | 2 |
| | | – All pumps are in the auto position | 0.121 | 3 |
| | | – Maintenance is carried out every three months and recorded | 0.113 | 4 |
| | | – The specific water related to the fire system is maintained and monitored daily, and the data is recorded | 0.105 | 5 |
| | | – The annual maintenance contract is signed with an approved maintenance contractor | 0.078 | 6 |
| | | – The diesel pump is tested weekly | 0.067 | 7 |
| | | – A certificate of compliance is issued annually | 0.051 | 8 |
| | | – The battery is tested periodically | 0.050 | 9 |
| | | – Residential buildings are insured | 0.039 | 10 |
| Fire accident management | 0.166 | – Fire accidents, near misses, and minor fire accidents are monitored, recorded, and analyzed | 0.296 | 1 |
| | | – Fire accidents, near misses, and minor fire accidents are monitored, recorded, analyzed, investigated, and reported to relative authorities, and the report is shared with residents | 0.220 | 2 |
| | | – Fire accidents, near misses, and minor fire accidents are monitored, recorded, analyzed, and investigated | 0.200 | 3 |
| | | – Fire accidents, near misses, and minor fire accidents are monitored, recorded, analyzed, investigated, and reported to the relevant authorities | 0.146 | 4 |
| | | – Fire accidents, near misses, and minor fire accidents are monitored, recorded, analyzed, investigated, and reported to the relevant authorities, and the report is shared with residents and the public | 0.139 | 5 |

**Table 2.** *Cont.*

| Criteria | Weight | Sub-Criteria | Weight | Ranking |
|---|---|---|---|---|
| Fire alarm system | 0.128 | – Fire alarm system is working, and the fire panel is energized 24/ | 0.357 | 1 |
| | | – Fire alarm system is working, and the fire panel is energized 24/7 without faults or fake fires | 0.241 | 2 |
| | | – Fire alarm system is working, and the fire panel is energized 24/7 without faults or fake fires, and it is connected 24/7 to an Aman early warning system | 0.222 | 3 |
| | | – Fire alarm connectivity with the 24/7 Aman early warning system is tested every three months | 0.181 | 4 |
| Fire extinguisher | 0.076 | – The fire extinguisher has been fixed according to the requirements in the UAE Fire and Life Safety Code of Practice | 0.702 | 1 |
| | | – The fire extinguisher has been fixed according to the requirements in the UAE Fire and Life Safety Code of Practice, and it has been maintained periodically and the corresponding data has been recorded | 0.298 | 2 |
| Employees | 0.068 | – All employees working in the residential building have attended approved firefighting training | 0.718376 | 1 |
| | | – All employees working in the residential building have attended approved firefighting training and will attend refresher training every two years | 0.281622 | 2 |
| Residents | 0.063 | – Resident data has been collected and includes the phone number, quantity, language, nationalities, ages for the purpose of risk assessments, and awareness of the residents | 0.273 | 1 |
| | | – IR 4.0 technology is used to provide fire awareness programming for all residents in their respective languages | 0.190 | 2 |
| | | – Systematic awareness programming is provided for all residents in their respective languages | 0.162 | 3 |
| | | – Fire awareness programming is provided for all newcomers | 0.115 | 4 |
| | | – Fire drills are carried out every 3 months by the attendant civil defense organization | 0.102 | 5 |
| | | – Resident monitoring system counting is performed | 0.082 | 6 |
| | | – Emergency layout contact numbers are fixed inside each flat and in all corridors | 0.077 | 7 |
| Service rooms, lifts, and gas connections | 0.046 | – Gas system sensors are installed in all kitchens and connected to the fire alarm system | 0.403 | 1 |
| | | – Lifts are connected to the fire alarm system | 0.310 | 2 |
| | | – Services rooms are free from fire hazards | 0.287 | 3 |
| Waste, housekeeping, and evacuation facilities | 0.043 | – Waste is removed daily (once or several times) according to the amount of garbage | 0.205 | 1 |
| | | – Exits routes are free from obstruction | 0.189 | 2 |
| | | – Housekeeping is carried out properly daily (once or several times) according to the needs of the building | 0.169 | 3 |
| | | – There is no parking within 15 m of the residential building | 0.155 | 4 |
| | | – Exit signs are fixed according to the requirements in the UAE Fire and Life Safety Code of Practice, and they are maintained periodically, and this data is recorded | 0.154 | 5 |
| | | – Assembly points are fixed according to the requirements in the UAE Fire and Life Safety Code of Practice | 0.128 | 6 |

Each sub-criterium holds a specific weight in the fire management system for high-rise buildings, and the weighting reflect the effect of the sub-criterium upon the overall fire measures. The calculated compliance ratings of sub-criteria reflect the levels of compliance with the fire indexes, according to the percentages shown in Table 3. HRBFI sub-criteria.

**Table 3.** HRBFI sub-criteria.

| | Sub-Criteria | Weight (%) |
|---|---|---|
| 1.1 | Competent fire risk professional is in place | 8% |
| 1.2 | Fire risk assessment has been carried out | 6% |
| 1.3 | The control of fire risk assessment is effective, monitored, and reviewed | 4% |
| 1.4 | A fire safety management system has been implemented | 4% |
| 2.1 | All suppression systems are connected to a 24/7 Aman early warning system | 4% |
| 2.2 | All pumps are energized 24/7 | 3% |
| 2.3 | All pumps are in the auto position | 2% |
| 2.4 | Maintenance is carried out every three months and is recorded | 2% |
| 2.5 | The specific water related to the fire system is maintained and monitored daily, and this data is recorded | 2% |
| 2.6 | The annual maintenance contract has been signed with an approved maintenance contractor | 1% |
| 2.7 | The diesel pump is tested weekly | 1% |
| 2.8 | A certificate of compliance is issued annually | 1% |
| 2.9 | The battery is tested periodically | 1% |
| 2.10 | Residential buildings are insured | 1% |
| 3.1 | Fire accidents, near misses, and minor fire accidents are monitored, recorded, and analyzed | 5% |
| 3.2 | Fire accidents, near misses, and minor fire accidents are monitored, recorded, analyzed, investigated, and reported to the relevant authorities, and the report is shared with residents | 4% |
| 3.3 | Fire accidents, near misses, and minor fire accidents are monitored, recorded, analyzed, and investigated | 3% |
| 3.4 | Fire accidents, near misses, and minor fire accidents are monitored, recorded, analyzed, investigated, and reported to the relevant authorities | 2% |
| 3.5 | Fire accidents, near misses, and minor fire accidents are monitored, recorded, analyzed, investigated, and reported to the relevant authorities, and the report is shared with residents and the public | 2% |
| 4.1 | Fire alarm systems are working, and the fire panel is energized 24/7 | 5% |
| 4.2 | Fire alarms system are working, and the fire panel is energized 24/7 without faults or fake fires | 3% |
| 4.3 | Fire alarm systems are working, and the fire panel is energized 24/7 without faults or fake fires, and they are connected to a 24/7 Aman early warning system. | 3% |
| 4.4 | Fire alarm connectivity is tested with the 24/7 Aman early warning system every three months | 2% |
| 5.1 | The fire extinguishers are fixed according to the requirements in the UAE Fire and Life Safety Code of Practice | 5% |
| 5.2 | The fire extinguishers are fixed according to the requirements in the UAE Fire and Life Safety Code of Practice, and they are maintained periodically and this data is recorded | 2% |
| 6.1 | All employees working in the residential building have attended approved firefighting training | 5% |
| 6.2 | All employees working in the residential building have attended approved firefighting training, and they will attend refresher training every two years | 2% |
| 7.1 | Resident data has been collected, including the phone numbers, quantity, languages, nationalities, ages for the purpose of risk assessments, and the awareness of the residents | 2% |
| 7.2 | IR 4.0 technology is used to provide fire awareness programming for all residents in their respective languages | 1% |
| 7.3 | Systematic awareness programming is provided for all residents in their respective languages | 1% |
| 7.4 | Fire awareness programming is provided for all newcomers | 1% |
| 7.5 | Fire drills are carried every 3 months by the attending civil defense authority | 1% |
| 7.6 | Resident monitoring system counting is performed | 0.5% |
| 7.7 | Emergency layout contact numbers are fixed inside each flat and in all corridors | 0.5% |
| 8.1 | Gas system sensors are installed in all kitchens and connected to the fire alarm system | 2% |
| 8.2 | Lifts are connected to the fire alarm system | 1% |
| 8.3 | Services rooms are free of fire hazards | 1% |
| 9.1 | Waste is removed daily (once or several times) according to the amount of garbage | 1% |
| 9.2 | Exit routes are free of obstructions | 1% |
| 9.3 | Housekeeping is carried out properly daily (once or several times) according to the needs of the building | 1% |
| 9.4 | There is no parking within 15 m of a residential building | 1% |
| 9.5 | Exit signs are fixed according to the requirements in the UAE Fire and Life Safety Code of Practice, and they are maintained periodically and this data is recorded | 1% |
| 9.6 | Assembly points are fixed according to the requirements in the UAE Fire and Life Safety Code of Practice | 1% |

Residential buildings are classified into three categories. According to the UAE ministerial resolution 505 from the year 2012, a category one residential building is one with a height of 23 to 46 m, or one with 7 to 15 floors; category two is a residential building with height of 46 to 90 m, or one with 16 to 30 floors; and category three is a residential building with a height of 90 m, or one with more than 31 floors.

To identify the minimum percentages to be achieved for each category of high-rise buildings, the criticality of each sub-criterium was evaluated using the Delphi technique through a subject matter expert that identified the minimum requirements for each category, as shown in Table 4.

**Table 4.** Minimum requirements index.

| | Sub-Criteria | Weight (%) | Cat-1 | Cat-2 | Cat-3 |
|---|---|---|---|---|---|
| 1.1 | Competent fire risk professional is in place | 8% | | √ | √ |
| 1.2 | Fire risk assessment has been carried out | 6% | √ | √ | √ |
| 1.3 | The control of fire risk assessment is effective, monitored, and reviewed | 5% | √ | √ | √ |
| 1.4 | A fire safety management system has been implemented | 4% | | √ | √ |
| 2.1 | All suppression systems are connected to a 24/7 Aman early warning system | 4% | √ | √ | √ |
| 2.2 | All pumps are energized 24/7 | 3% | √ | √ | √ |
| 2.3 | All pumps are in the auto position | 2% | √ | √ | √ |
| 2.4 | Maintenance is carried out every three months and this data is recorded | 2% | √ | √ | √ |
| 2.5 | The specific water related to the fire system is maintained and monitored daily, and this data is recorded | 2% | √ | √ | √ |
| 2.6 | An annual maintenance contract has been signed with an approved maintenance contractor | 1% | √ | √ | √ |
| 2.7 | The diesel pump is tested weekly | 1% | √ | √ | √ |
| 2.8 | A certificate of compliance is issued annually | 1% | √ | √ | √ |
| 2.9 | The battery is tested periodically | 1% | √ | √ | √ |
| 2.10 | Residential buildings are insured | 1% | √ | √ | √ |
| 3.1 | Fire accidents, near misses, and minor fire accidents are monitored, recorded, and analyzed | 5% | √ | √ | √ |
| 3.2 | Fire accidents, near misses, and minor fire accidents are monitored, recorded, analyzed, and investigated | 3% | √ | √ | √ |
| 3.3 | Fire accidents, near misses, and minor fire accidents are monitored, recorded, analyzed, investigated, and reported to the relevant authorities. | 2% | √ | √ | √ |
| 3.4 | Fire accidents, near misses, and minor fire accidents are monitored, recorded, analyzed, investigated, and reported to the relevant authorities, and the report is shared with residents | 4% | | √ | √ |
| 3.5 | Fire accidents, near misses, and minor fire accidents are monitored, recorded, analyzed, investigated, and reported to the relevant authorities, and the report is shared with residents and the public | 2% | | | √ |
| 4.1 | Fire alarm systems are working, and the fire panel is energized 24/7 | 5% | √ | √ | √ |
| 4.2 | Fire alarm systems are working, and the fire panel is energized 24/7 without faults or fake fires | 3% | √ | √ | √ |
| 4.3 | Fire alarm systems are working, and the fire panel is energized 24/7 without faults or fake fires and they are connected to a 24/7 Aman early warning system. | 3% | √ | √ | √ |
| 4.4 | Fire alarm connectivity is tested with the 24/7 Aman early warning system every three months | 2% | | | √ |
| 5.1 | The fire extinguishers are fixed according to the requirements in the UAE Fire and Life Safety Code of Practice | 5% | √ | √ | √ |
| 5.2 | The fire extinguishers are fixed according to the requirements in the UAE Fire and Life Safety Code of Practice, and they are maintained periodically and this data is recorded | 2% | √ | √ | √ |

**Table 4.** *Cont.*

| | Sub-Criteria | Weight (%) | Cat-1 | Cat-2 | Cat-3 |
|---|---|---|---|---|---|
| 6.1 | All employees working in the residential building have attended approved firefighting training | 5% | √ | √ | √ |
| 6.2 | All employees working in the residential building have attended approved firefighting training, and they will attend refresher training every two years | 2% | √ | √ | √ |
| 7.1 | Resident data has been collected, including phone numbers, quantity, languages, nationalities, ages for the purpose of risk assessments, and awareness of the residents | 2% | | √ | √ |
| 7.2 | IR 4.0 technology is used to provide fire awareness programming for all residents in their respective languages | 1% | | | √ |
| 7.3 | Systematic awareness programming is provided for all residents in their respective languages | 1% | | √ | √ |
| 7.4 | Fire awareness programming is provided for all newcomers | 1% | | √ | √ |
| 7.5 | Fire drills are carried out every 3 months by the attending civil defense authority | 1% | | √ | √ |
| 7.6 | Resident monitoring system counting is performed | 0.5% | | | √ |
| 7.7 | Emergency layout contact numbers are fixed inside each flat and in all corridors | 0.5% | √ | √ | √ |
| 8.1 | Gas system sensors are installed in all kitchens, and they are connected to the fire alarm system | 2% | √ | √ | √ |
| 8.2 | Lifts are connected to the fire alarm system | 1% | √ | √ | √ |
| 8.3 | Service rooms are free of fire hazards | 1% | √ | √ | √ |
| 9.1 | Waste is removed daily (once or several times) according to the amount of garbage | 1% | | √ | √ |
| 9.2 | Exit routes are free of obstructions | 1% | √ | √ | √ |
| 9.3 | Housekeeping is carried out properly daily (once or several times) according to the needs of the building | 1% | √ | √ | √ |
| 9.4 | There is no parking within 15 m of the residential building | 1% | | √ | √ |
| 9.5 | Exit signs are fixed according to the requirements in the UAE Fire and Life Safety Code of Practice, and they are maintained periodically and this data is recorded | 1% | √ | √ | √ |
| 9.6 | Assembly points are fixed according to the requirements in the UAE Fire and Life Safety Code of Practice | 1% | √ | √ | √ |
| | | | 72% | 95% | 100% |

The high-rise residential buildings that achieve a score of below the minimum points for each category are considered high risk because the fire protection requires improvements to be capable of fighting fires, and the possibility of fire accidents occurring remains high. A residential building that achieves the minimum points is considered to have met the minimum requirements for fire protection. A residential building that meets more than the minimum requirements is considered to have proper fire procedures.

By using the technologies of the Fourth Industrial Revolution, the Sharjah Civil Defense Authority (SCDA) can provide distinguished, accurate, and fast services based on the data collected by the early warning system Aman, which can be used to perform a self-examination of the fire and alarm systems in residential buildings.

Based on the system's data and other criteria, it is possible to provide electronic and self-compliance certificate issuance services. The system can also perform self-monitoring procedures to measure the level of compliance of residential buildings with fire requirements and take actions to inform the building's management electronically, without human intervention in the reporting process. Digitalizing all other services provided by the SCDA could have a significant impact if the technologies of the Fourth Industrial Revolution, such as artificial intelligence, robotics, big data, and the Internet of Things, are used.

Based on the collected electronically data related to the high-rise building fire index and the efficiency of response of each building, the SCDA can decide the level of compliance, which will vary from one building to another, according to the geographical locations and heights of the buildings. The data taken electronically from the Sharjah Electricity, Water, and Gas Authority will provide a proper estimation of the number of residents in each building, which will add value to the fire prevention procedures.

Emergency response may be applied to reduce the residual risk of escalation and to obtain an improved prevention of fire escalation. Emergency response can reduce accident losses, and previous studies have evidenced its role in preventing domino effects [36]. The Minnesota fire department regulation definition of fire response is: any deployment of firefighting personnel and/or equipment to extinguish a fire or perform any preventative measure in an effort to protect equipment, life, or property in an area threatened by fire. It also includes the deployment of firefighting personnel and/or equipment to provide fire suppression, rescue, extrication, or any other services related to fire and rescue as may occasionally occur. Emergency response is restricted by many factors, and emergency resources (including emergency personnel) are important factors influencing emergency response efficiency, influencing even whether an emergency response can be carried out. The quantity, scheduling, and allocation of emergency resources may influence an emergency response. Emergency resource allocation arranges the resources required for an emergency response in order to deal with unexpected events so that the emergency response process can be carried out efficiently [37]. Building fire emergencies are considered a high-risk domain for the significant loss of lives and property. Accurate information and situational awareness (SA) enable fire responders to make timely decisions and perform safe operations during fire emergencies [38]. Manoj and Baker stated that technological issues could cause fatalities in emergencies. The critical issues are related to a lack of critical information sharing, a lack of communication systems, and a lack of decision support systems [39].

The last firewall for fighting fire comes from the competent authority for implementing rescue and fire extinguishing operations. The UAE Fire and Life Safety Code of Practice specifies a period of eight minutes, which is the distance between any center of the Authority Civil Defense and the location of the fire. Of course, this time period can be less, depending on the time of the accident and traffic, but the eight-minute period will not be enough to control the fire without control measures before the arrival of the firefighting teams, and so the validity of the fire alarm and fire systems and the training of residents and workers is important.

The response to fire accidents is critical to a fire management system's framework. A fire accident is considered a failure of fire prevention and fire protection, which is concluded to be a fire accident. The failure may be a lack of training or a failure of the firefighting systems, fire alarm systems, awareness, or training, and the immediate action to control such a failure is to respond to a fire accident in time to control the fire before it affects the lives of people or damages the building. The framework implemented in the Emirate of Sharjah depends partially on the digital response, and the automatic response to a fire decreases the duration of the time required to reach the fire. It provides accurate information about the residential building, as well as information such as the number of stories and occupants, an accurate location, and the story of the fire. The response to a fire will be effective if it is classified according to the type of building, the number of floors, the area of the building, the access to the building, the fire protection index, the time of the fire accident, and whether it is day or night.

*3.1. Response Effectiveness*

A response will be affected by the percentage achieved in the high-rise building index. If the building has less than the minimum points, the situations described below in Table 5 may take place.

**Table 5.** HRBFI response failure results.

| Criteria | Weight | | Failure Results |
|---|---|---|---|
| Fire risk assessment | 22% | – | Without fire risk control, the hazard is not identified and not controlled |
| Fire suppression system | 18% | –<br>– | The system failed, and the water tank capacity was below the required volume to extinguish the fire<br>The fire pumps are not working |
| Fire accident management | 16% | – | The cause of the accident may have been unknown at the time of the response due to a lack of data |
| Fire alarm System | 13% | – | The cause of the accident may have been unknown at the time of the response due to a lack of data |
| Fire extinguishers | 7% | – | The fire extinguishers may not have been maintained and may not have been serviceable for use |
| Employees | 7% | – | Employees not trained may play negative roles during a response |
| Residents | 7% | – | Residents are not aware of the response procedures |
| Service rooms, lifts, and gas connections | 4% | –<br>–<br>– | Fire may burned the insides of the service rooms, which may affect the electricity and water or ignite gas<br>Some residents may try to use the lifts<br>Gas systems that are not linked to a fire alarm system will be turned on, and during a fire, this is more dangerous |
| Waste, housekeeping, and response facilities | 6% | –<br>–<br>– | If the exit signs are not lit, the residents may fail to reach the exit routes<br>Waste and garbage will be fuel for the fire<br>The parking-free area around the building is less than 15 m, which obstructs the rescue efforts |

To measure the effectiveness of a response and rescue, the calculation of several variables according to the high-rise building fire index is required through the use of the analytic hierarchy process (AHP). These calculations are to include the floor number and the distance from the nearest fire station. Table 6. HRBFI response details.

**Table 6.** HRBFI response details.

| Criteria | Weight | Absolute Error |
|---|---|---|
| CR = 0.01 | | |
| High-rise building fire index | 0.699 | 0.024 |
| Distance from fire station | 0.153 | 0.005 |
| High-rise building category | 0.148 | 0.005 |

According to the distance from the fire station to the location of a residential buildings, each single kilometer affects the response rate by 1%, up to a maximum distance of 15 km from a fire station.

According to the UAE fire and life fire code and the three HRB categories, for category 3, the percentage will be a maximum of 15%, as shown in Table 6. HRBFI response details for category one, the percentage will be one-third of the weight, and for category two, it will be calculated as two-thirds of the weight. For category 3, the weight will be calculated as a full 15%, as shown in Table 7.

**Table 7.** Response calculation sample.

| HRBFI | | | Floors | | | Distance 1 km = 1% | Response Efficiency (%) |
|---|---|---|---|---|---|---|---|
| | | | Cat (1) | Cat (2) | Cat (3) | | |
| Sample | SCORE | 70% | 15% | 10% | 5% | 15% | |
| Sample (1) | 100% | 70% | 15% | 0 | 0 | 15% | 100% |
| Sample (2) | 100% | 70% | 0 | 10% | 0 | 14% | 94% |
| Sample (3) | 100% | 70% | 0 | 0 | 5% | 13% | 88% |

An automatic fire response decreases response time, provides enough information to make the right decision, improves the quality of the response, decreases the possibility of injury and fatality, decreases the time required to extinguish the fire, and decreases the possible damage to the building or the assets. The response tools, such as trucks, equipment, and communication systems, should be prepared to support the digital response and use the technology as effectively as possible to improve the quality of the response.

The above information will be of added value to the civil defense authority for making decisions when receiving a fire report, as shown in Figure 2. The automatic response will allow the decision-maker to view the data and make the right decision related to rescue and response procedures, or the system can make a decision by itself, according to saved data.

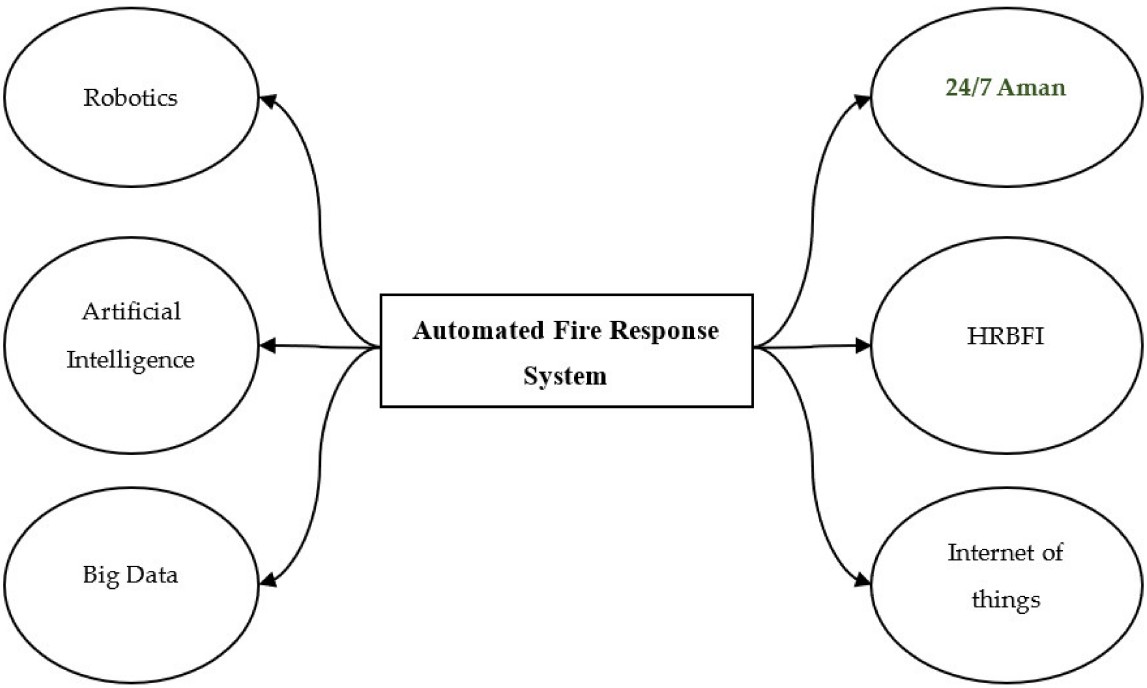

**Figure 2.** Automated fire response system.

*3.2. The Early Warning System 24/7 Aman*

It is necessary to develop a scientific and reasonable method to create an on-site early warning about a fire in order to facilitate wise decisions for the evacuation of firefighters at the most appropriate time, which may greatly enhance the efficiency of the firefighters' rescue [40]. Serious fire accidents happen frequently every year worldwide, and they kill many people and result in significant property loss. The sensitive fire alarm systems (FASs) that can detect an early fire in a timely manner and immediately provide an alarm are, therefore, urgently needed [41]. A rapid, accurate, and early warning detection system for fires has been shown to be a very effective method for minimizing casualties and property damage, especially in buildings [42].

The 24/7 Aman early warning system provides a real-time reporting mechanism for fire incidents and works to provide accurate data on the geographical location of a

residential building to facilitate access as soon as possible. It also provides data on the number of floors in the building and the number of residents, which helps civil defense officials make appropriate decisions and work out a response that is appropriate to the size of the fire, as well as to select the appropriate rescue equipment and determine the required number of firefighters.

### 3.3. HRBFI Compliance

Compliance is based on the HRBFI, which provides data on a building's compliance with the requirements of fire legislation. Residential buildings with low compliance rates must expect complex rescue and firefighting operations where all the necessary precautions must be taken for the success of the response. Buildings that achieve a high compliance rate have preventive measures that will help with the success of the response and fire control.

### 3.4. Internet of Things

The Internet of Things (IoT) is an organization comprised of interconnected gadgets. These gadgets are equipped for detecting their current circumstance and sharing and handling information that can be made accessible to an assortment of utilizations [43]. The Internet of Things (IoT) is a term that is frequently used in the current digital era. It is defined as a system of integrated or interconnected sensor nodes/actuators that receives, measures, generates, publishes, and intelligently shares information or details among other sensor nodes [44].

The operations of controlling fire and alarm systems can be self-controlled by taking advantage of the Internet of Things, which will enables such systems to perform initial response procedures without human intervention and control fire systems, partial and complete evacuation, and energy sources, while counting the number of people and predicting the people inside the building who were not evacuated, as shown in Figure 3.

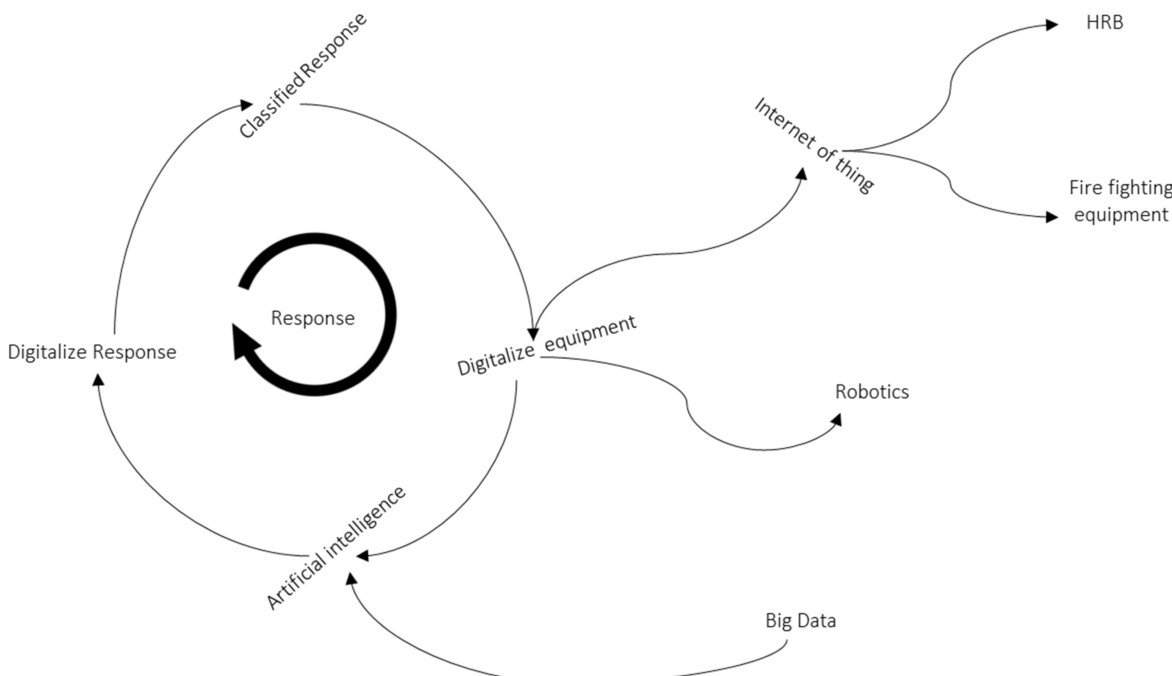

**Figure 3.** Response flowchart.

### 3.5. Robots

Robots can also play a vital role in evacuation operations. They can reach areas that are not accessible to humans during a fire, and they can self-control to evacuate injured people and carry out complex extinguishing operations, which will reduce human losses and improve the efficiency of a response.

### 3.6. Artificial Intelligence

Artificial Intelligence is one of the key research techniques that several researchers have analyzed, and it has proven to be the best at improving the performance of detecting fire hazards in smart cities [45],

Artificial Intelligence, or machine learning techniques, can contribute to predicting accidents before they occur, and they can predict the movement of fire in a building and determine the core of a flame to combat it. Artificial Intelligence can also contribute to identifying people who are not likely to evacuate or places that could be dangerous for firefighters to enter.

### 3.7. Big Data

With the development of digital technology, big data has a significant impact on the operation, cooperation, and strategies of different fields in the digital economy. Big data refers to real-time, large amounts of structured, semi-structured, and unstructured data which are converted into value by specific technologies and analytical methods [46].

Big data can provide important information for predicting or controlling fire accidents. This data can be collected from social media, from devices used in a residential building, or from mobile phones, all of which can provide important data that contributes to the success of a response, improving comfort and reducing losses as much as possible.

An efficient response depends on the modernized technology used, and modern equipment directly contributes to increased response efficiency in a reinforcing loop. A classified response improves response efficiency by gathering information and clarifying the resources needed to make the right decision at the time of the response. IR 4.0 technologies can increase response efficiency, as it considers the critical elements in a response when relying directly on reflection, making the response more accurate and easier, while relying on machines rather than on humans. Using big data, Artificial Intelligence, and the Internet of Things, it is possible to reduce errors, increasing response efficiency and contributing to minimizing possible losses in a balanced loop, as shown in Figure 4.

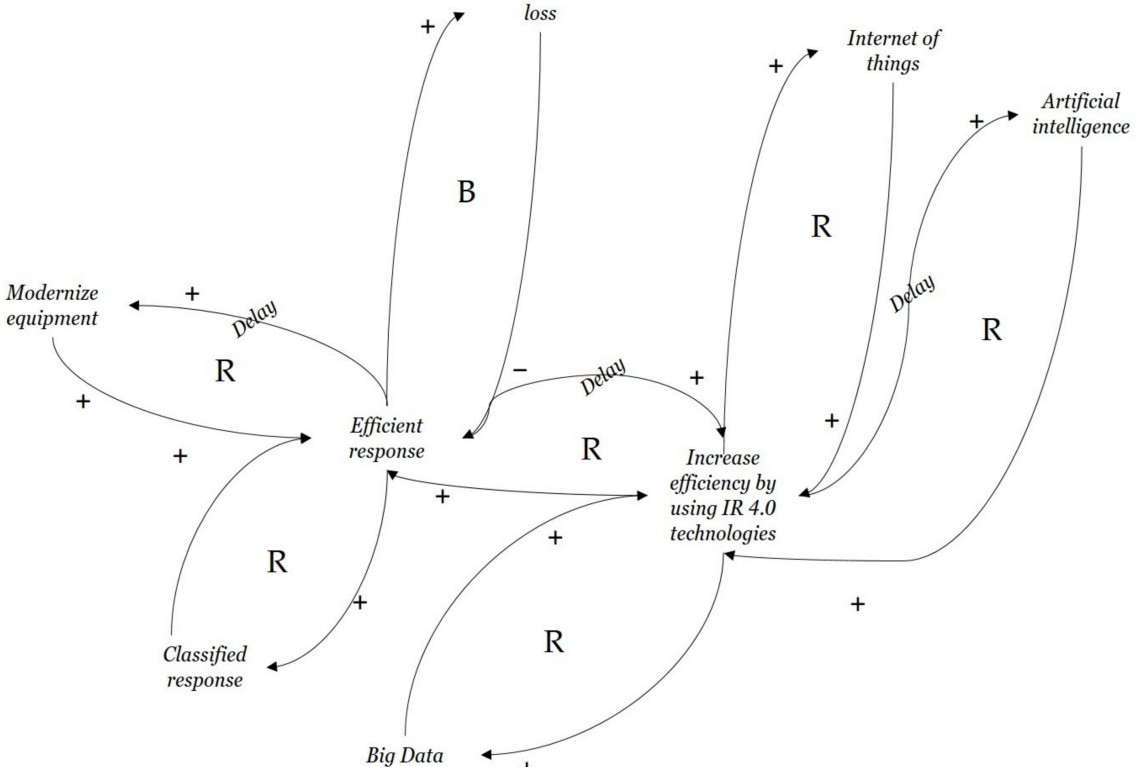

**Figure 4.** Response efficiency.

## 4. Conclusions

This study proposes a fire index for high-rise building compliance in the Emirate of Sharjah. A proper literature review was carried out to identify the fire factors affecting residential buildings, and a comprehensive analysis was carried out on the factors related to high-rise buildings in the Emirate of Sharjah. The primary criteria were identified, and using multiple research tools and consulting multiple subject matter experts, a fire index was developed to measure high-rise building compliance based on the circumstances in the Emirate of Sharjah, which will contribute to improving the level of compliance with the fire requirements.

Based on the index we developed to measure fire requirements compliance, another index was developed to measure the efficiency of a fire response. This second index was designed according to the level of compliance, the distance to the nearest fire station, and the height of the building in terms of floor numbers. This study linked the two indexes, which could increase both compliance and the efficiency of responses.

**Author Contributions:** M.O.: writing—original draft, visualization, investigation, and data curation. A.M.: supervision, methodology, and conceptualization. S.B.A.A.: review and validation. All authors have read and agreed to the published version of the manuscript.

**Funding:** This research received no external funding.

**Institutional Review Board Statement:** Not applicable.

**Informed Consent Statement:** This study did not involve humans.

**Data Availability Statement:** The data presented in this study are available on reasonable request from the corresponding author.

**Conflicts of Interest:** The authors declare no conflict of interest.

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
