# Peer review of "Fire Safety Index for High-Rise Buildings in the Emirate of Sharjah, UAE"

_fire, doi:10.3390/fire6020051_

Round 1

Reviewer 1 Report

There is a lack of research justification. Need to justify the research needs.  Section 1 (Introduction) and 3 (literature review) can be merged under the introduction section and then research methodology needs to write properly. The authors did not discuss about the method of literature review. What databases are used to carry out the literature review. How many journal/conference papers are used? How was the compliance index developed? How the index system for measuring the fire response efficiency was developed, need to justify? What’s methods were adopted?

1.     Who are stakeholders? It would be better to discuss about their expertise.

 2.     What is IR 4.0? It would be better to introduce first.

3.     What is HRBs? It would be better to introduce first.

4.     How the index system for measuring the fire response efficiency was developed, need to justify? What’s methods were adopted?

5.     “…….the wait for building….” should be “…….the weight for building….”

6.     “……….fire extinguisher by 5 pints……” should be “……….fire extinguisher by 5 points……”

7.     How the point 5, 15 and 30 was selected, need to justify?

8.     In Table 2, there are serial number 2.1…2.11, 3.1….etc, but serial number 7 is missing?

9.     Conclusion is not clear. Need to separate first two sentences. It would better to summarise the key conclusions in dot points.

10.  For peer-review journal articles, number of references need to increase more than 30.

Author Response

Dear: reviewer 

Thanks for your valuable comments on the manuscript. I worked to go through the comments one by one, which made a big change to the content of the paper. I returned to the subject matter experts again because I made changes to the research methodology to remove the notes.

 already sent the document to the MDPI English editing services for editing waiting for them.

Reviewer 2 Report

Please see comments below: - 

No in-text citation in the introduction section

The methods are not well detailed and explained. What methodological approach was adopted for this study? Was it quantitative, qualitative or mixed methods? What was the research design, approach, data collection, data analysis technique etc. These are all missing. 

Literature review is not thorough and does not explicitly provide critical analysis of published literature on this topic.

Grammatical errors can be found throughout the manuscript

Author Response

Dear: reviewer 

Thanks for your valuable comments on the manuscript. I worked to go through the comments one by one, which made a big change to the content of the paper. I returned to the subject matter experts again because I made changes to the research methodology to improve the quality of the paper.

 I already sent the document to the MDPI English editing services for editing waiting for them to finalise the edititng langaue. 

Round 2

Reviewer 1 Report

The authors tried their best to improve the paper. Can be accepted with proper English language checking.